# Genetic Variations and Haplotypic Diversity in the Myostatin Gene of Different Cattle Breeds in Russia

**DOI:** 10.3390/ani11102810

**Published:** 2021-09-27

**Authors:** Elena Konovalova, Olga Romanenkova, Anna Zimina, Valeria Volkova, Alexander Sermyagin

**Affiliations:** L.K. Ernst Federal Research Center for Animal Husbandry, Dubrovitsy 60, Podolsk Municipal District, 142132 Moscow, Russia; ksilosa@gmail.com (O.R.); filipchenko-90@mail.ru (A.Z.); moonlit_elf@mail.ru (V.V.); alex_sermyagin85@mail.ru (A.S.)

**Keywords:** beef cattle, myostatin gene, mutations, *nt821(del11)*, double-muscling, *F94L* polymorphism, Aberdeen Angus, Limousin, Simmental, Belgian Blue breed

## Abstract

**Simple Summary:**

This paper presents the results of the study of two polymorphisms of the myostatin gene associated with muscular hypertrophy in the Russian populations of Aberdeen Angus, Limousin, Simmental, and Belgian Blue cattle breeds. For their diagnostics, test systems based on modern molecular genetic methods were developed, and the population analysis showed a low frequency of the undesirable allele associated with the genetic defect of double-muscling, and a high frequency of the allele that presumably positively influences meat productivity traits.

**Abstract:**

The myostatin gene (*MSTN*) in cattle has a number of polymorphisms associated with increased muscle mass. The aim of the current study was to determine the haplotype frequencies of *F94L* and *nt821(del11) MSTN* polymorphisms among cattle bred for meat in Russia, using DNA analysis. Using the earlier created test systems based on the AS-PCR and PCR-RFLP methods, six populations of Aberdeen Angus (*n* = 684), two populations of Limousin (*n* = 54), one population of Simmental (*n* = 55), and one population of Belgian Blue (*n* = 137) belonging to Russian farms were genotyped on *nt821(del11)* and *F94L*
*MSTN* polymorphisms. The animal carriers of the mutant allele of *nt821(del11)*
*MSTN* associated with the double-muscling genetic defect were found in one Aberdeen Angus population at a frequency of 2.18%, but were not found in the Limousin and Simmental populations. However, 100% of the Belgian Blue population were heterozygous carriers of *nt821(del11)*
*MSTN*. The frequencies of the *A* allele *F94L*
*MSTN* desirable for productivity traits in the Limousin populations were the highest and accounted for 0.97 and 1 in populations one and two, while in the Aberdeen Angus, Simmental, and Belgian Blue populations, these figures were considerably lower at 0.04–0.08, depending on the population. The obtained data show the high genetic potential of Russian beef cattle, and facilitate an improvement in meat productivity by preserving the health of animals.

## 1. Introduction

Myostatin protein was initially a subject of research interest, due the appearance of animals with significantly high muscle mass. This observation led to the coining of the term “muscular hypertrophy”. The first written account of bovine muscular hypertrophy (mh) was by a British farmer in his stock almanac [1]. This was later described in some depth in 1888 by Kaiser [2], who changed the perception of the meat characteristics of cattle, marking the beginning of a novel perspective on animal breeding with increased muscle mass. This phenotype has since become increasingly widespread among European beef cattle and has been termed “double-muscling” (DM). It has occurred at a high frequency in some breeds of cattle, such as Belgian Blue and Piedmontese [3]. To differentiate between DM and the normal phenotype among animals, researchers have used different symbols. These include double-muscled or normal; DM or N; D or n; DM or dm; C or N; c or n; A or a; and mh or + [4].

In addition, the influence of the myostatin gene on bone development has been established: newborn myostatin null mice (MSTN−/−) possess larger vertebral bone volume fractions (bone volume/total volume; BV/TV) due to increased BV, trabecular thickness (Tb, Th), and bone area (BA), as well as increased tissue mineral density (TMD) and BMD, as compared to newborn wildtype (Wt) mice [5].

Interestingly, the molecular structure of the myostatin protein is conserved across vertebrates so that its C-terminal sequences are 100% identical in mice, rats, humans, pigs, and chickens [6].

Myostatin, or growth and differentiation factor 8 (*GDF8*), was initially identified as the factor causing a double-muscling phenotype due the presence of mutations inactivating gene, and, therefore, leading to the loss of the ability to stop muscle fiber growth [6]. The genetic study of the myostatin gene (*MSTN*) began during the last century [7,8].

The influence of *MSTN* variability on the productivity traits of agriculturally useful animals has been investigated in different species. In particular, the polymorphisms of the promoter region of the *GDF8* gene influence meat color and final pH, affecting meat quality in various sheep breed groups [9]. PCR with single-stranded conformational polymorphism (PCR-SSCP) analysis of the intron-1 of sheep (*Ovis aries*) was used to identify five variants, designated *A–E* (Gene Bank No. FJ858196-FJ858200) of the *MSTN* gene [10]. General linear mixed effect models revealed that the presence of *A* allele in a lamb’s genotype was associated with decreased leg, loin, and total yield of lean meat, whereas the presence of *B* allele was associated with increased loin yield and proportion loin yield (loin yield divided by total yield expressed as percentage). The effect of the number of allele copies present was investigated, and it was found that the absence of *A* allele, or the presence of two copies of *B* allele, was associated with increased mean leg yield, loin yield, and total yield. Two copies of *B* allele were also associated with a decrease in the proportion of shoulder yield, whereas two copies of *A* allele were associated with a decrease in the proportion of loin yield [10].

The main reason for the appearance of mh is the gene mutations that result in a lack of *MSTN* expression and continuous muscle growth [11,12]. This was confirmed when the mh locus was mapped 3.1 cM from microsatellite TGLA44 on the centromeric end of bovine chromosome 2. The positional candidate approach carried out by Grobet et al. in 1997 demonstrated that a mutation in bovine *MSTN*, which encoded myostatin, a member of the TGFβ superfamily, is responsible for the double-muscled phenotype [13]. This was also reported on in regard to 11 bp deletion in the coding sequence for the bioactive carboxy-terminal domain of the protein, causing muscular hypertrophy observed in Belgian Blue cattle [14]. When the gene underlying the trait was identified as myostatin [12], it was found to have a surprisingly high number of polymorphisms [15].

Similar to other animal species, the myostatin gene in cattle is highly polymorphic [12]. To date, reports on several mutations in all coding regions of myostatin (exons 1–3) have described them as silent and causing non-synonymous changes. The most examined mutations of the *MSTN* gene mainly single nucleotide polymorphisms (SNP) appearing in the various cattle breeds are summarized in Table 1.

Phylogenetic analysis has shown that there was positive selection pressure for non-synonymous mutations within the myostatin gene family around the time of the divergence of cattle, sheep, and goats, and these positive selective pressures on non-ancestral myostatin are relatively recent [16]. Unfortunately, breed management problems exist for double-muscled cattle, such as birthing difficulties, which can be overcome through genetically controlled breeding programs [17].

We are interested in the polymorphisms of the myostatin gene caused by *nt821(del11)* and *F94L MSTN* mutations. In our opinion, these considerably influence the productivity traits of the *nt821(del11) MSTN* mutation associated with the double-muscled trait in cattle, which is characterized by an increase in muscle mass of approximately 20%, resulting in substantially higher meat yield, a higher proportion of expensive cuts of meat, and lean and very tender meat, for which a substantial premium is paid [17]. This trait is autosomal recessive, and the locus is referred to as mh. It occurs at such a high frequency in Piedmontese and Belgian Blue cattle that it is characteristic of these breeds [5]. However, it also occurs in other breeds (Table 1). In addition to its obvious advantages, double-muscling also has one major drawback—a greatly increased incidence of calving difficulties to the extent that cesarian sections are required for deliveries within these breeds [17] and so it was classified as a genetic defect (M1). However, due to its sufficient advantages, double-muscled cattle play a major role in animal agriculture in several countries, and they can be found in many regions of the world [18,19].

*F94L MSTN* is a single-nucleotide polymorphism (SNP) of g. 433C > A transversion, that causes amino acid substitution of phenylalanine via leucine at position 94 of the protein sequence. Previous studies have shown the additive effect of this mutation on cattle carcass traits and strong evidence that an *MSTN* genotype can produce an intermediate, non-double-muscling phenotype, which is expected to be of significant value for beef cattle producers [20,21].

Currently, the most precise methods of *nt821(del11)* and *F94L MSTN* diagnostics are DNA analyses, which are based on different modifications of the polymerase chain reaction (PCR) technique, and the most modern methods of DNA analyses are based on sequencing or real-time PCR [9,10,11,12,13,14,15,16].

The aim of this study is to determine the haplotype frequencies of the *F94L* and *nt821(del11)* polymorphisms of the myostatin gene among the Russian cattle populations of Aberdeen Angus, Limousin, Simmental, and Belgian Blue breeds using DNA analysis.

## 2. Materials and Methods

The study was approved by the bioethical commission of L.K. Ernst Federal Research Center for Animal Husbandry (Protocol No. 5 from 22 March 2021). We used the DNA samples (*n* = 930) from Aberdeen Angus, Simmental, Limousin, and Belgian Blue cattle (Table 2). The animals used in this study were selected from breeding farms located in different regions of Russia. All animals of the herds were semi-sibs and were managed and fed in the same manner.

The DNA were extracted from the available biomaterial of the animals (skin, sperm, milk, and blood) by methods traditionally used in the lab of genetics and genomics of cattle at L.K. Ernst Federal Research Center for Animal Husbandry (Dubrovitsy, Moscow, Russia) [22].

The genotyping methods were chosen based on the structures of the studied mutations. Due the presence of the small deletion for *nt821(del11)* genotyping, we used the allele-specific PCR method (AS-PCR) [22]. For *F94L* polymorphism diagnostics, we chose the PCR-RFLP (PCR with the subsequent analysis of restriction fragments length polymorphism) method due the existence of a mutation area recognition site for the *TaqI* restriction endonuclease site in the DNA.

For the development of the *nt821(del11)* and *F94L MSTN* genotyping tests, we chose the last version of the *Bos taurus* myostatin gene DNA sequence from NCBI (Accession Number MK 214682.1). The conducted alignment of the sequences using BLAST 2.12.0+ (https://blast.ncbi.nlm.nih.gov/Blast.cgi#1692837278, accessed on 20 August 2021) revealed high similarity between the chosen sequence and other *Bos taurus* myostatin gene sequences (99.46–100%).

The schemes of the test systems for *nt821(del11)* and *F94L MSTN* genotyping are presented in Figure 1.

Primer design was carried out using the Primer3Plus Software (https://primer3plus.com, accessed on 29 June 2021) and the selection of the restriction endonuclease was made using NEBcutter V.2.0. (https://nc2.neb.com/NEBcutter2/, accessed on 29 June 2021). The primer characteristics for the DNA amplification of the mutation area of *nt821(del11)* and *F94L MSTN* polymorphisms are presented in Table 3.

The PCRs were carried out on a Biorad T100 thermocycler (Biorad, Singapore) in a mixture of 10 μL M1 and 15 μL *F94L* containing dNTP mix, 10x PCR buffer, 0.03 μL of each oligonucleotide primer at a 100 mM concentration, and 1 U of *Taq*-polymerase. The volume of the DNA was 1 μL with the content of genomic DNA at 50 ng. The temperature–time PCR conditions included the initial denaturation under 95 °C for 3 min; 35 cycles of denaturation under 95 °C for 45 s; annealing under 60 °C (for M1) or 55 °C (for *F94L*) for 30 s; elongation under 72 °C for 30 s; and the final extension under 72 °C for 4 min.

The PCR-RFLP method was used to test the *F94L MSTN* polymorphism. Due to the observation of the restriction site for *TaqI* endonuclease restriction (T↓CGA) in the wild type but not in the mutant allele, we used this enzyme for RFLP analysis (1 U on a probe with exposition per night). PCR product detection was conducted by the method of electrophoresis in agarose gel with agarose content of 3% under a voltage of 120 V for 30 min. The 10x TAE buffer was used as a conducting medium.

Verification of the correct work of our test systems was carried out using the Sanger sequencing method. Sequencing was performed by Eurogene (Moscow, Russia). The results were evaluated using UGENE Software 40.0.

The statistical evaluation of the data was conducted using traditional methods [23,24] and with the assistance of Microsoft Office Excel and online calculators’ software (https://www.statskingdom.com, accessed on 28 June 2021). For the evaluation of the influence of breed factors on the frequencies of the desirable alleles of *nt821(del11)* and *F94L MSTN* mutations, we used two-way ANOVA (https://www.statskingdom.com/two-way-anova-calculator.html, accessed on 20 August 2021).

## 3. Results

The created DNA tests for the diagnostics M1 and *F94L* polymorphisms of *MSTN* allowed for the screening of the Russian populations of the examined meat cattle breeds.

### 3.1. DNA Tests for the Determination of the Allele Variants of M1 and F94L MSTN Polymorphisms

#### 3.1.1. *nt821(del11)* MSTN Polymorphism

As *nt821(del11)* polymorphism was an 11 bp deletion, the test system for the diagnostics of this mutation was based on the AS-PCR method, allowing the simultaneous amplification of wildtype and mutant alleles. For the implementation of the technique, three oligonucleotide primers were designed: one forward (MSTNF) for the amplification as a wildtype and as a mutant allele, and two reverse, one of them (MSTNRn) with the connection with the MSTNF-generated DNA fragment of 310 bp responding to the wildtype allele, and the other (MSTNRm) with the MSTNF-generated mutant allele of 144 bp. Therefore, the animals free from the mutation showed only one DNA fragment of 310 bp on gel electrophoresis; the animal carriers of the *nt821(del11)* mutation showed two DNA fragments of 310 and 144 bp on gel electrophoresis (Figure 2); and the animals homozygous on the *nt821(del11)* mutant allele, which in our investigation was not found, presented with only one DNA fragment of 144 bp on gel electrophoresis.

#### 3.1.2. *F94L MSTN* Polymorphism

As *F94L* polymorphism is the SNP, the test system for the diagnostics of this polymorphism was based on the PCR-RFLP method. The recognition site was found in the wildtype allele of *F94L MSTN* for *TaqI* restriction endonuclease. Therefore, after amplification and *TaqI* digestion, the wildtype allele representing the healthy animal was restricted on two DNA fragments of 118 and 92 bp. The mutant allele that does not enter the restriction site rested without changes and showed only the DNA fragment of 210 bp on gel electrophoresis. The heterozygous animals having in their genotypes wild type *C*-allele and mutant *A*-allele (*CA* genotype) showed three DNA fragments of 210, 118, and 92 bp on gel electrophoresis (Figure 3).

The sequencing of DNA amplicon of *nt821(del11) MSTN* genotyping is presented in Figure 4.

The sequencing of DNA amplicon of *F94L MSTN* genotyping is presented in Figure 5.

### 3.2. Results of Genotyping of the Russian Populations of Meat Cattle Breeds on nt821(del11) and F94L MSTN Polymorphisms

The genotyping of the Aberdeen Angus (populations 2–6, *n* = 638), Limousin, and Simmental cattle did not reveal the animal carriers of 11 bp deletion or known associations with double-muscling (M1) genetic defect (*M1F* animals). The *M1C* animals were detected in Aberdeen Angus population one at a frequency of 2.18%, and the average of its presence in all investigated populations of this breed was 0.15 ± 0.07%. In contrast, all cows in the Belgian Blue population were carriers of *nt821(del11) MSTN* mutation associated with double-muscling. The genotyping results of all investigated populations regarding the two *MSTN* gene polymorphisms are shown in Table 4.

The genotyping of the *F94L MSTN* polymorphism showed that most animals of the studied cattle populations of the Aberdeen Angus, Simmental, and Belgian Blue breeds had the *CC* genotype. The percentage of animals with this genotype was 91.58–96.36% depending on the population. The heterozygous animals with the *CA* genotype were found in Aberdeen Angus cattle populations at a frequency of 6.90% (population one) and 5.60% (population six). They were also observed in Limousin population one at a frequency of 3.33%. Most animals of the Limousin breed were carriers of the *AA* genotype, which is desirable in terms of productivity. In Limousin population one, this figure was 96.67%, and in population two, 100% of animals had the *AA* genotype. The *CC* genotype in the Limousin populations was not found.

The frequencies of the desirable *A* allele were the highest in the Limousin populations (0.97 and 1.0 in populations one and two, respectively). In the studied populations of the Aberdeen Angus, Simmental, and Belgian Blue breeds, the number of *AA* animals was relatively low at 0.04–0.08.

The conducted two-way ANOVA showed that the mutation (factor A) did not influence the breed (factor B) (this is a logical outcome in our opinion), and the *p*-value was 0.0217037 (*p*(*x* ≤ 9.477369) = 0.978296). This suggests that the chance of a type I error (rejecting a correct H_0_) is small: 0.0217 (2.17%). The smaller the *p*-value, the more it supports H_1_. For factor B (breed), the *p*-value was 0.122585 (*p*(*x* ≤ 3.039146) = 0.877415). This suggests that the chance of a type I error (rejecting a correct H_0_) is too high: 0.1226 (12.26%). The larger the *p*-value, the more it supports H_0_. Thus, we could not exclude the influence of breed on the haplotypic diversity of the populations relating to *nt821(del11)* and *F94L MSTN* polymorphisms.

In Figure 6, the numbers of the animals with frequencies of the desirable wildtype allele of *nt821(del11)* and *A-F94L* polymorphisms among the studied cattle populations of the Aberdeen Angus, Limousin, and Simmental breeds are demonstrated. It can be clearly observed that Limousin breed populations seem to be the most beneficial in regard to the studied myostatin mutations.

## 4. Discussion

The conducted investigations showed the sufficient variability of the myostatin gene and revealed almost all genotypes of *nt821(del11)* and *F94L MSTN* mutations in the cattle populations of Aberdeen Angus, Limousin, Simmental, and Belgian Blue.

Previous investigations of *nt821(del11) MSTN* in Belgian Blue cattle have shown that all heads of this breed were carriers of the mutant allele associated with double-muscling [25]. The additional *nt821(del11) MSTN* genotyping conducted in the frame of the current work has confirmed the earlier obtained data and conclusion of our colleagues overseas, that the mutant allele associated with the double-muscling genetic defect was fixed in the Belgian Blue cattle breed due the long-time selection in regard to increased muscle mass [26]. On the one hand, this polymorphism has a positive influence on meat quality by increasing fat content in carcasses [26], but, on the other hand, *nt821(del11)* in the homozygous station is a reason behind the calving difficulties or dystocia ultimately leading to the wide use of the cesarian section in Belgian Blue cattle [19,27].

The *F94L MSTN* polymorphism occurred due to the considerable interest in the single-nucleotide polymorphism c.282C > A (or g.433C > A) showed by breeders. In fact, this mutation has the same action as that of *nt821(del11)* on fat content, but it does not have an influence on the ease of calving due the absence of influence on birth weight, thereby making this polymorphism a possible genetic marker of productivity traits [28].

Previous investigations have shown the nonsignificant effect of the *A* allele of *F94L MSTN* on birth and growth traits, and intramuscular fat content in carcasses of Limousin backcross calves in Australian and New Zealand populations. However, a significant increase in total carcass fat weight (−16.5% and −8.1% trait means for Australian and New Zealand populations; *p* < 0.001 and *p* < 0.05, respectively), meat weight (7.3% and 5.9% trait means for Australian and New Zealand populations, respectively, *p* < 0.001), and a reduction in fat depth (−18.7%, *p* < 0.001 in the Australian population) have been established. Meat tenderness, pH, and cooking loss of *M. longissimus dorsi* were not affected by the *F94L* variant. The results offer strong evidence that this *F94L MSTN* variant provides an intermediate and more useful phenotype than the more severe double-muscling phenotype caused by knockout mutations in the myostatin gene [20]. In accordance with other data, the g.433A allele was found to be associated with a 5.5% increase in silverside percentage and eye muscle area (EMA), and a 2.3% increase in total meat percentage relative to the g.433C allele. The phenotypic effects of the g.433A allele were partially recessive, providing strong evidence that an *MSTN* genotype can produce an intermediate, non-double-muscling phenotype, which is expected to be of significant value for beef cattle producers [21].

A population analysis previously conducted on fifteen cattle breeds (*n* = 1140) located in Australia demonstrated the presence of SNP 433C > A in the Simmental (0.8%), Piedmontese (2%), Droughtmaster (4%), and Limousin (94.2%) breeds, but not in the Salers, Aberdeen Angus, Hereford, Charole, Jersey, Holstein, Shorthorn, Gelbvieh, and Maine-Anjou breeds [20].

In the current study, we obtained data that are consistent with the data previously obtained by other researchers [28], according to which the most genetically preferable breed in terms of *F94L MSTN* polymorphism is the Limousin breed. The data on the Simmental breed are also in accordance with those of an Australian study—0.04% *AA* animals in the Russian population and 0.08% in the Australian one. Regarding the Aberdeen Angus breed, unlike in the Australian populations, in our herds, SNP 433C > A was found in *CA* animals with frequencies of 6.90% and 5.60% in Aberdeen Angus populations one and two, respectively. It is also noteworthy that if among the populations of the Aberdeen Angus breed, the animals that are carrying the A allele are completely heterozygous (*CA* genotype), then the animals of the Simmental and Belgian Blue breed populations that have the *A* allele in their genotype are homozygous for this allele.

In general, the populations we studied are characterized by a low proportion of heterozygosity for both of the studied *MSTN* polymorphisms. *F94L MSTN* was observed in one of the populations of the Limousin breed as well as in the populations of the Simmental and Belgian breeds, but this was not the case for animals with the *CA* genotype. Regarding *nt821(del11) MSTN* polymorphism, the heterozygous animals (M1C or *MSTN+/MSTN*−) were absent from almost all investigated populations, excluding the Belgian Blue breed. These facts are probably related both to the purebred breeding practiced in the farms that owned the studied populations, and to the historical origin of the studied cattle breeds. Currently, among these indicators, we can observe a fairly high stability of the genome of the Russian cattle of the Aberdeen Angus, Simmental, Limousin, and Belgian Blue breeds in relation to the studied mutations of the myostatin gene.

The obtained results also allow for the proposal of the high genetic potential of the Limousin and Belgian Blue cattle breeds. In our opinion, the Belgian Blue cattle could be used in selection programs for the purpose of productivity improvement, but with caution due to the high risk of the appearance of reproductive problems. The Limousin breed is extremely promising, due to the high frequency of the polymorphism linked with the increase in muscle weight but without calving difficulties. The high frequencies of *A* allele *F94L MSTN* polymorphism may be the result of a directed, long-term selection of animals with a high muscle mass combined with the simultaneous culling of animals with the appearance of calving difficulties.

Despite the low frequency of *AA* genotype animals in the populations of the Aberdeen Angus and Simmental cattle breeds, the presence of animals carrying the *A* allele genotype represents the possibility of cattle selection with consideration of *F94L MSTN* polymorphism. In addition, we cannot discount the good acclimatization abilities and endurance of these animals as a result of this—they are already spread widely over the territory of Russia and form the breeding core of beef cattle breeding.

The results of the current study show the high genetic potential of Russian Aberdeen Angus and Limousin livestock.

In our opinion, the issue of polymorphisms of the myostatin gene has not yet been thoroughly examined. In 2012–2013, four double-muscled German Gelbvieh cattle were born with an abnormally high birth weight of more than 60 kg, and specialists proposed that this effect could be explained by either the homozygous c.821del11 mutation or compound heterozygosity of the c.191T > C and c.821del11 mutations [29].

In addition, we should not forget the prospective programs of genome selection, which allow one to select the most productive animals based on the knowledge of information regarding polymorphism of all genomes. In 2013, the reference population of 3060 beef cattle, including pure-bred Limousin, Limousin cross-bred with Angus, and pure-bred Angus, were genotyped using a BovineSNP50 BeadChip and directly for the *MSTN*-*F94L* variant. The study demonstrated that the *MSTN*-*F94L* variant was the most strongly associated with five traits (birth weight, easy and direct calving, milk, weaning weight, and yield grade) among the thirteen measured traits in the Limousin and mixed (Angus x Limousin) populations. Fitting the *MSTN*-*F94L* variant as a random effect, the genomic prediction accuracies for birth weight increased by 2.7% in purebred Limousin, by 2.2% in the mixed populations, and by 3.2% in the multibreed population. Prediction accuracies for five traits increased in the multibreed analysis. Fitting *MSTN-F94L* as a fixed effect in purebred Limousin, Angus x Limousin, and multibreed analyses resulted in increased prediction accuracy in purebred Limousin for eight traits. Prediction accuracies can be improved by including a causal variant in genomic evaluation rather than simply using whole-genome SNP markers [30].

It should be noted that recent research findings in humans and other mammalian and non-mammalian species support the potent regulatory role of myostatin in the morphology and function of muscle, as well as cellular differentiation and metabolism, with real-life implications in agricultural meat production and human disease. Myostatin-null mice (*MSTN*−/−) exhibit skeletal muscle fiber hyperplasia and hypertrophy, whereas myostatin deficiency in larger mammals, such as sheep and pigs, can result in muscle fiber hyperplasia. Myostatin’s impact extends beyond muscles, with alterations in myostatin present in the pathophysiology of myocardial infarctions, inflammation, insulin resistance, diabetes, aging, cancer cachexia, and musculoskeletal disease [31].

## 5. Conclusions

The results of the investigations conducted in this study allow us to conclude that the myostatin gene is extremely variable and not unambiguous because in this gene, there are polymorphisms that have a positive effect on meat productivity traits as well as those that have negative effects on their reproductive traits, for example, calving difficulties (dystocia). The genotyping results of Russian populations of Aberdeen Angus, Limousin, Simmental, and Belgian Blue cattle show that the domestic herds of these breeds could be considered as beneficial for further breeding. The most investigated populations had low frequencies of animal carriers of the double-muscling genetic defect caused by *nt821(del11)* mutation of the myostatin gene. The exception was the Belgian Blue cattle population, which consisted of 100% animal carriers of the mutant allele generated due the 11 bp deletion and associated with the M1 genetic defect (double-muscling).

The study of *F94L MSTN* polymorphism revealed the high frequency of the desirable A allele in terms of productivity in the Limousin populations, making this cattle breed attractive in terms of an improvement in the meat productivity without negative consequences related to animal health.

The Aberdeen Angus cattle populations showed the presence of *A* allele *F94L* and the mutant allele of *nt821(del11) MSTN* (although at low frequencies), which suggests that it may be possible to develop their meat qualities, facilitating the production of beef with a larger weight and lower fat depth in carcasses.

Thus, the studied populations of cattle of the Aberdeen Angus, Limousin, Simmental, and Belgian Blue breeds showed a sufficiently high genetic potential, suggesting the possibility of increasing the productivity of herds while preserving their health.

Currently, searches are being carried out to identify a method of obtaining animals with knock-out myostatin using CRISPR/CAS9 technology [32]. However, in reality, the selection of animals based on genotyping information seems to be the most affordable way to increase the productivity of beef breeds of cattle.

## Figures and Tables

**Figure 1 animals-11-02810-f001:**
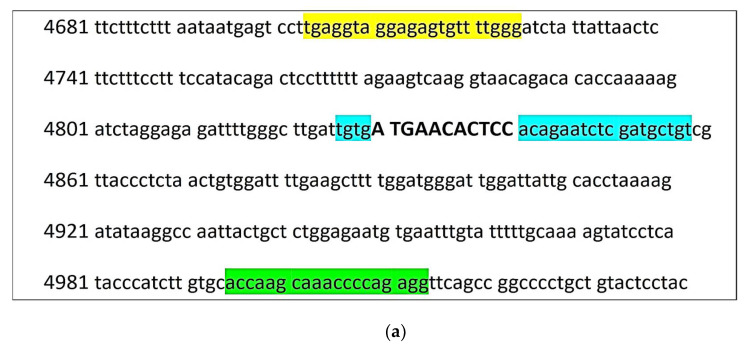
Schemes of DNA tests for *MSTN* polymorphisms genotyping. (**a**) Scheme of *nt821(del11) MSTN* genotyping test system: in yellow—forward primer (MSTNF), in green—reverse primer for the amplification of the wildtype allele (MSNTRn), in blue—reverse primer for the amplification of the mutant allele (MSTNRm). The 11 bp deletion is indicated by the capital letters in bold; (**b**) Scheme *of F94L MSTN* genotyping test system: in yellow—forward primer (F94LF), in green—reverse primer (F94LR). Restriction site for *TaqI* endonuclease is indicated by arrow.

**Figure 2 animals-11-02810-f002:**
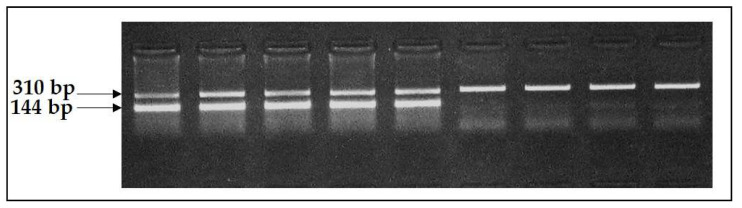
Electropherogram of genotyping for M1 *MSTN* mutation. The wildtype allele is the upper fragment of 310 bp and the mutant allele is the lower fragment of 144 bp. The first five lines correspond to the animal carriers of *nt821(del11) MSTN* mutation and the further four lines to the animals free from the mutation.

**Figure 3 animals-11-02810-f003:**
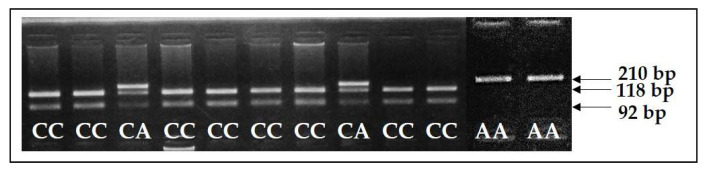
Electropherogram of *F94L MSTN* genotyping. Wildtype allele *C* was restricted with the generation of two DNA fragments of 118 and 92 bp. Mutant allele *A* was not restricted and presented only one DNA fragment of 210 bp.

**Figure 4 animals-11-02810-f004:**
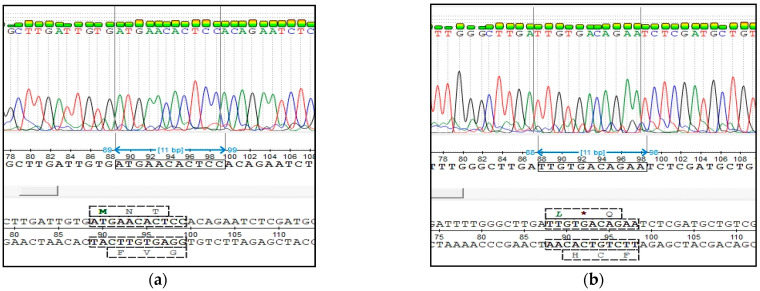
Results of the Sanger sequencing of PCR products of *nt821(del11) MSTN* genotyping: (**a**) animal free from the mutation (M1F); (**b**) animal carriers of the mutation (M1C).

**Figure 5 animals-11-02810-f005:**
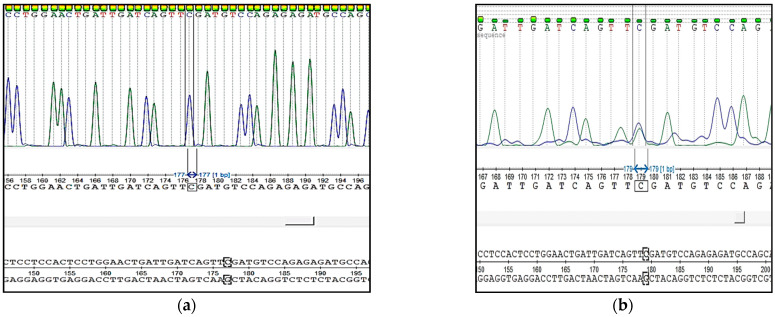
Results of the Sanger sequencing of PCR products of *F94L MSTN* genotyping: (**a**) animal free from the mutation (*CC* genotype); (**b**) heterozygous animal (*CA* genotype).

**Figure 6 animals-11-02810-f006:**
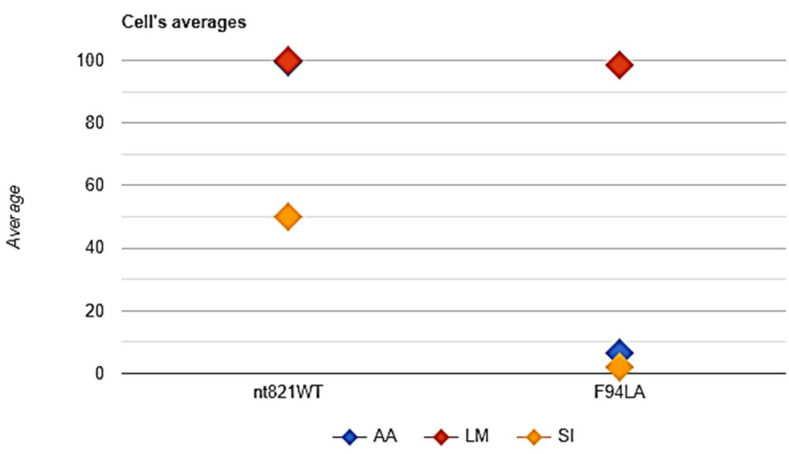
Image of the two-way ANOVA of the studied populations of Aberdeen Angus (AA), Limousin (LM), and Simmental (SI) cattle breeds. Nt821WT—wildtype allele of *nt821(del11)* polymorphism, F94LA—*A* allele of *F94L MSTN* polymorphism.

**Table 1 animals-11-02810-t001:** Short description of the most examined *MSTN* mutations in bovine [3,13,14].

SNP *	Mutation	Amino Acid Change	Breed
nt821(del11)	c.821–831del1(tgaacactcca)	p.Glu275ArgfsX14	Asturiana, Belgian Blue, Blonde d’ Aquitaine, Limousin, Parthenaise, South Devon, Santa Gertrudis, Braford, Murray Grey, Angus
nt267	A > G	Silent mutation	Aubrac, Bazadaise, Salers
nt324	C > T	Silent mutation	Asturiana
nt387	G > A	Silent mutation	Asturiana, Salers, Galloway
C313Y	c.938G > A	p.Cyc313Tyr	Gasconne, Piedmontese, Parthenaise
E226X	c.610G > T	p.Glu226X	Maine-Anjou, Marchigiana
E291X	c.871G > T	p.Glu291X	Maine-Anjou, Marchigiana
F94L	c.282C > A	p.Phe94Leu	Angus, Limousin
Q204X	c.610C > T	p.Gln204X	Blonde d’Aquitaine, Charolaise, Limousin
S105C	c.314C > G	p.Ser105Cys	Parthenaise
L64P	c.191T > C	p.Leu64Pro	German Gelbvieh, Glanrind, Limpurger
D182N	c.544G > A	p.D182N	Asturiana

* Single nucleotide polymorphism.

**Table 2 animals-11-02810-t002:** Description of research material.

Breed	Population	Gender	*n*
Aberdeen Angus	1	Bulls	46
	2	Bulls	37
	3	Bulls	40
	4	Mixed (bulls and heifers)	73
	5	Bulls	296
	6	Cows	192
Limousin	1	Cows	34
	2	Heifers	20
Simmental	1	Bulls	55
Belgian Blue	1	Cows	137

**Table 3 animals-11-02810-t003:** Primers’ sequences and PCR conditions for genotyping on *nt821(del11)* and *F94L MSTN* mutations.

Primer Name	Primer Sequence (5′–3′)	Position in the Gene	Product Length (Base Pairs)
MSTNF	tgaggtaggagagtgttttggg	c.4704–4725	310 (MSTNF + MSTNRn) 144 (MSTNF + MSTNRm)
MSTNRn	cctctggggtttgcttggt	c.4995–5013	
MSTNRm	acagcatcgagattctgtcaca	c.4826–4858	
F94L	tgagaacagcgagcagaagg	c.224–243	210
F94R	actccgtgggcatggtaatg	c.506–525	

**Table 4 animals-11-02810-t004:** Results of genotyping of Russian populations of Aberdeen Angus, Limousin, Simmental, and Belgian Blue cattle in regard to *nt821(del11)* and *F94L MSTN* polymorphisms.

Breed	Population	*nt821(del11)* (%)		*F94L* Genotype (%)			*F94L* Allele	
		*M1F*	*M1C*	*CC*	*CA*	*AA*	*C*	*A*
Aberdeen Angus (*n* = 238)	1	97.82	2.18	93.10	6.90	0.00	0.93	0.07
	6	100.00	0.00	94.44	5.60	0.00	0.94	0.06
Limousin (*n* = 54)	1	100.00	0.00	0.00	3.33	96.67	0.03	0.97
	2	100.00	0.00	0.00	0.00	100	0.00	1.00
Simmental (*n* = 55)	1	100.00	0.00	96.36	0.00	3.64	0.96	0.04
Belgian Blue (*n* = 137)	1	0.00	100.00	91.58	0.00	8.42	0.92	0.08

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
