# Peer review of "Genetic Variations and Haplotypic Diversity in the Myostatin Gene of Different Cattle Breeds in Russia"

_animals, 2021, doi:10.3390/ani11102810_

Round 1

Reviewer 1 Report

The authors described the results of genotyping 4 cattle breeds in Russia on F94L and nt821del11 mutations in the MSTN gene. 

The work is understandable to the reader, and the following chapters are described in a manner appropriate for this type of research. My comments are minor.

The introduction is written correctly and in my opinion, is sufficient to understand the subject of research. In the introduction, you can mention other species in which mutations in the GDF9 gene have been associated with increased meatiness. 

Materials and Methods.

In the case of animals from one population if you take unrelated animals for the study?

Table 3. I would rather move PCR conditions into the text and add some information about product length and the name of primers into the table.

The results and discussion are properly and clearly describe.

Author Response

Dear reviewer,

Thank you very much for the kindly review of our manuscript.

I would like to clarify that the object of our study was the polymorphisms of GDF8 (not GDF9) gene. But we agree with the comment and input the information with the respective references about the influence of the GDF8 polymorphisms on ovine productive traits.

In the case of animals from one population, the animals are belonged to the one herd from one farm. Due to the using of artificial insemination technology on the farms and the existence of the strategy which allows in some cases for the improvement of the productivity of animals the related crossbreeding, we can`t exclude some relation degree between the animals. We can propose that the animals of the herds were half-sibs.

In according to your wishes, the PCR conditions have been moved into the text and in the table 3 the PCR products lengths have been added.

Thank you again for the review.

With the best wishes, the team of authors.

Reviewer 2 Report

The study shows the results of analysis of two myostatin gene polymorphisms in the Russian populations of Aberdeen Angus, Limousine, Simmental and Belgian Blue cattle breeds. For their diagnostics the DNA test based on AS-PCR and RFLP- PCR was used; two molecular methods not so modern, today small deletions, as well as SNP mutations, can be easily analyzed by PCR +capillary eletrophoresis (fragment analyses and Snpshot), this technique allowing the identification of all genotypes.

The analysis of the association with the morphological traits has not been carried out, this could give more original to the work on the Russian populations, the frequencies in the breeds are already known. In addition, the study has serious problems, such as the RFLP results interpretation. For this I thinks that the article cannot be published in this form but must be thoroughly reviewed, the discussion must be rewritten.

Here are some suggestions for the revision of your work:

In Grobet 1997 (https://www.ucm.es/data/cont/docs/345-2013-11-08-myostatin_double_muscled_cattle.pdf) is reported the molecular structure of the deletion nt821del11 , insert it in the introduction.

in the materials and methods…

It is not clear if the PCR protocols used in this work are the same used in  reference 19, if so then it is better to say it (line 98-100)
The position of the primers used is not clear: report the reference sequence used to draw them and the position with respect to deletion (nt821del11), you can reported a graphic representation of gene sequence region. You report expected sizes of fragments
The nt821del11 protocol it is not clear how the three genotypes are distinguished with AS-PCR: how do you distinguish the heterozygous from the homozygous with deletion?

The procedure used for fragment sequencing was not reported.

Line 120-129 allelic and genotypic frequency formulas and mean population could be removed, are basic concepts of genetics
in Results…
Why are only heterozygous subjects shown in Figure 1? Must be reported all three genotypes (as for F94L MSTN, figure 2) to illustrate the result of the typing method and especially the specificity of the AS-PCR: the primer Mstm must amplify only the allele with deletion, this has been verified? How?

In Figure 1 the sequence shown does not appear to be the deletion region: the 11 bases are different from reported in the text and in Grobet 1997. It would be better to report the alignment of the sequences obtained with the reference sequence.

Figure 4 : all 3 genotype must be reported, it would be better the sequences alignment.

Table 3 : in Aberdeen Angus population, it seems that results are reported only of the populations 1 and 6, why are not reported the others?
Figure 5 shows the same data as Table 3

3.1.2. F94L MSTN polymorphism (Line 147-155)
There seems to be a misinterpretation of the RFLP results: the checked mutation is g433C>A, so the wildtype allele is C, mutant A. The enzyme used recognizes the TTCGAT sequence, then it cuts the allele C (wild type) while the allele A (TTAGAT sequence) is not cut (210bp) This would justify the differing results reported in discussion.

Discussion

Line 198-200 it is not clear: what do you mean whit haplotype?

Line 236: is the reference 14 correct? Maybe is it reference 15

Author Response

Dear Reviewer,

Thank you very much for the attention to the manuscript so detailed review of it.

We agree that the AS-PCR and PCR-RFLP are not most modern methods and don`t reject from the using of more advanced molecular techniques such real-time or digital PCR. But we consider AS-PCR and PCR-RFLP methods as the initial link for the development of any molecular diagnostic test system. In addition, these traditional methods are the most accessible and least dependent on the qualification of the laboratory assistant and computer software, which allows to obtain the results of genotyping in any molecular genetic laboratory. 

We also agree that the correlation analysis between the different allele variants of the studied polymorphisms and some productive traits would considerably increase the value of our work and improved the manuscript. Unfortunately, at the moment, we do not have data on the productivity of the studied populations. But, getting the message about the preparing the Special issue on Myostatin gene we considered it our duty to submit all the information received in the course of our work, as it can be useful for breeders of cattle of the Aberdeen-Angus, Limousine, Belgian blue and Simmental breeds. Also, we cannot fully agree that the frequencies in the breeds are already known. In the different geographic regions, there are different approaches to the cattle breeding, caused by the different needs of the people population, that can`t don’t influence on the genetic structure of the cattle populations.

The PCR protocols used in this work is only partially the same used in the earlier published our work (on reference in Archives Animal breeding journal). It`s a test system for the nt821del11 genotyping. The respective reference has been input in the text.

Also, the more detailed explanation of the work of test system for the nt821del11 diagnostics was added in the text.

The Sanger sequencing of the DNA fragments has been conducted under the helping of Eurogene Company (Moscow, Russia) which we also mentioned in the manuscript.

We have also changed the Figure 1 on the analogue one with the animals free from the nt821del11 mutation, that additionally demonstrated the ability of this test system to distinguish the wild-type and the mutant alleles. Due the animals homozygous on the mutant allele of nt821del11 were absent in the studied population, the third genotype mh/mh has not been present.

You quite rightly noticed that the primer that the primer MSTNRm must amplify only the allele with deletion. Firstly, it was designed in according to the DNA sequence of MSTN and in second, the Sanger sequencing conducting with the primer has confirmed the amplification of the mutant allele with deleted 11 bp.

About the comment that in Figure 1 the sequence shown does not appear to be the deletion region: the 11 bases are different from reported in the text and in Grobet 1997.  It should be noted that in Figure 1 present the results of gel electrophoresis of nt821del11 MSTN genotyping showing only the strips of the amplified DNA fragments. Probably, you mean Figure 3 presented the Sanger sequencing results of nt821del11 genotyping. We would like to clarify that on the left part of the Figure (a) the 11 bp was marked in the square and they the same as ones from the earlier reported by Grobet 1997 (ATGAACACTCC). This part of the Figure responds to the animal free from the mutation and so the presence of these 11 bp is certainly. On the right part of Figure 3 (b) the mutant allele generated due the 11 bp deletion has been presented. And, respectively, the 11 bp sequence (ATGAACACTCC) in this case is absent. The alignment of the sequencies will be done and the primers positions have been pointed graphically.

We also must explain the data presented in the Table 3. In fact, all six Aberdeen Angus populations have been investigated on nt821del11 polymorphism and only in population 1 the animals-carriers M1 have been observed. The populations Nos. 2-6 consisted from 100% M1 free animals (in the text these data have been pointed). But on F94L polymorphism only populations Nos. 1 and 6 have been investigated.   So, for the saving space on the page, we decided to present the data in this manner.

We can`t agree that Figure 5 shows the same data as Table 3. In Table 3 have been presented the GENOTYPE frequencies of the studied polymorphisms allowing to conclude about the genetic diversity of the populations relating to the nt821del11 and F94L MSTN. The Figure 5 showed the ALLELE frequencies of F94L MSTN polymorphism that allows to evaluate the populations from the point of view of their genetic potential.

I would like to say a special thank you to the reviewer for noticing a misinterpretation of the RFLP results of F94L polymorphism genotyping. It was indeed a gross mistake negatively influenced as on quality of our work as on the obtained results. It was corrected as in the all parts of the manuscript and the results, conclusions and discussion were also revised.

In the chapter of Discussion, the meaning of haplotype has been changed on genotype. And the references have also been revised and in some cases corrected. 

Thank you again and the best regards,

the team of authors.

Reviewer 3 Report

The manuscript from Konovalova et al. aims at identifying polymorphisms in the Myostatin (MSTN)  with muscle growth in various Russian cattle breeds.
Using a AS-PCR and PCR-RFLP approach and subsequent sequencing of PCR fragments, the author describe n821del11 and F94L MSTN polymorphisms.

Generally, the manuscript is very poor. This paper has several weaknesses and needs improvement before publication. This manuscript has major language problems. There are too many for me to understand them all. Authors are strongly encouraged to seek a native English speaker who may assist you modifying the document.

1. The title is windy and convoluted. It needs streamlining to make it self-explanatory without ambiguity. I suggest changing it to something along the lines of “ Genetic variations and haplotypic diversity in the Myostatin gene of different Russian Cattle breeds

1. In the Introduction focus on the objectives and insert a few new reference and relevant findings

2. Material and method needs to clarifying and summarizing- some detailed needs
3. The subtitles in the material and method needs to summarizing Ethical approval and references must be mentioned in M&M

4.The conclusion is also weak and incoherent. What is the new information from your finding that has been added to already existing knowledge on MSTN?

Specific comments:

Line 28 : What does it mean? Meat cattle? rephrase the whole abstract

Table 1 : How about Kambadur et al., 1997, McPherron and Lee, 1997, Grobet et al., 1997, Grisolia et al., 2009). 

Line 59 : Which mutations?

Line 78 : Reference?

Line 64: Doesn't make sense! Why you've chose to study these mutations? 

Line 68 :Reference

Line 70 : Irrelevant

Line 78: Reference?

Line 85: Why the most ? Base on what?

Lind 86: PCR-RFLP?

Lind 93: Animal ethics approval number must be included

Table 2 : How were the animals selected?

Lind 98 -100: Why two different PCR approach were chosen for two mutations?

Line 109:  Specify approximate amount of DNA in the aliquot

Table 3: What is the third primer in the table stand for ? Forward or reverse?

Lind 115: Specify how many amplicons per pattern were sequenced.

Line 116 : in TBE buffer?

Line 117 : Did you test for statistical differences between breeds? Specify what tests were undertaken. Also, the results from these need to be presented. 

Line 134 to 138: Rephrase it. 

Line 159 -160: Move to M & M

Line 178 : Table 4 ??  Include number of animals (n=) genotyped in each group, in brackets next to the breed name. Alternatively, you could specify the actual number of animals (in brackets) next to each genotype frequency.

Discussion section is poorly written – needs significant improvement.

Please check the nomenclature of MSTN  SNPs. You need to check and make sure you use the correct symbols for the SNPs that you refer to. You should use the same system consistently.

Author Response

Dear Reviewer,

Thank you for your attention to our manuscript and for the useful comments.

In according your comment, we have change title of the manuscript on the more understandable for the auditory of the journal.

The Material and method have been clarified and summarized with the inserting of some additional details.

The information about the Ethical approval has been mentioned in M&M chapter.

The conclusion as the whole manuscript has been modified. We consider as a new information from our work the obtaining the new knowledge of the genetic structure of the Russian populations of beef cattle breeds relating to myostatin gene what would give a possibility to use this information as the domestic as foreign breeders. In addition, we share the used methods, what also can be useful for the employees of molecular genetic laboratories because the main advantages of our developed techniques are their simplicity and accessibility.     

About the specific comments.

The phrase “meat cattle” was changed on more concrete “beef cattle breeds”.

When listing mutations of the myostatin gene in the text, references were given to the authors who first reported them.

We would like to clarify that in the Introduction we have used the term of non-synonymous mutations. A nonsynonymous mutation is a nucleotide mutation that alters the amino acid sequence of a protein. In contrast them are the synonymous substitutions (sometimes silent mutations), which do not alter amino acid sequences. As nonsynonymous substitutions result in a biological change in the organism, they are subject to natural selection.

The need references have been also imputed in the text.

We also realize that the methods used of us are not the most modern. But the primary goal was to evaluate the genetic structure of the Russian populations of beef cattle breeds and these methods turned out to be the most accessible in our circumstances. At about the same time, an offer was received to contribute to a special issue of the journal Animals dedicated to myostatin. So we decided to submit to the journal the obtained results. In future, of course, we are planning to modern the developmental test-system by fluorescent detection methods. In the Introduction we have specified the most modern methods of DNA analysis with the relevant links.

Explanations were also given about the selection of animals, the choice of research methods, the amount of DNA required for PCR tests. The principle of work of the AS-PCR-based test system for genotyping samples using the nt821(del11) MSTN polymorphism was explained in more detail, in particular, that MSTNF primer – forward common for as the wild-type as a mutant allele, and MSTNRn and MSTNRm primers are revers, solving for the amplification of the wild-type and mutant alleles, respectively.

The PCR amplicon sequencing has been conducted by order of Eurogene Company (Moscow, Russia) specialists what we also mentioned in the manuscript.

For the gel electrophoresis we used tris-acetate buffer (TAE) in 10x dilution.

In according to your comment, we have also conducted two-ways ANOVA. The reference on the respective software has been imputed in the M&M and the results into the relevant chapter.

In Table 4 we have pointed the quantity of investigated animals below to the breed’s name.

The Discussion has also been revised and we hope improved.

The symbols of the gene polymorphism have been correct in according to the internationally accepted designations.

Thank you again for your review,

Best regards.

The team of authors.

Round 2

Reviewer 2 Report

The work is thoroughly revised and all suggestions have been accepted; the current form is clear and complete.

Author Response

Dear reviewer,

thank you very much for your kind attention to our manuscript and for the contribution that you have made to its improvement. We believe that your comments considerably improved the quality of the article and hope and it will worthy of publication in the Animals journal.

Reviewer 3 Report

The paper is very badly written. It seems that was good research conducted for this paper, but the use of English and how it is presented makes it
really hard to understand anything. I recommend giving the paper to a native
speaker and work with them to make it understandable.

comments:

Line 73-77: References?

Line 77: What is variant A and B stand for? I see still some inconsistency in the variants name . please use the varnomen.hgvs.org to report the description of sequence variants.

Line 92- Other species? rephrase the sentence!

Line 112: why productivity trait ?

Figures 1 and 2 : poor resolution and also not necessary. Delete?

Lind 401: Can you explain the low frequency of heterozygous in your studied populations ? It is interesting that no homozygous AA found in Aberdeen Breed and no CC found in Limousine!

Please move the results of figure 7 to table 4! not necessary to separate them.

Line 685: How you claim that?

Author Response

Dear reviewer,

Thank you very much for your patience and attention to our manuscript.

Let us clarify some of the comments.

In according to your recommend we gave the paper to a native English speaker and hope the done corrective work made it understandable.

Lines 73-77 were devoted to the description of allelic variants of the myostatin gene and their influence on the signs of sheep productivity. For a better understanding of the issue, we have added some information about the method by which five alleles of the myostatin gene intron-1 polymorphism (DNA fragment 414 bp) were identified, as well as the numbers of these alleles in the NCBI gene bank. Relevant links have also been added.

Line 92- The word combination “Other species” was rephrased in “Other animal species”.

In the manuscript we have used the concept of productivity traits.  Under the productivity traits we understand live weight, slaughter yield, the ratio of individual varietal cuts in the carcass, the composition of meat and its nutritional value, that is, everything that can affect the final output of beef and its cost.

Figures 1 and 2 were deleted but we have changed Figure 1 to reflect primers` sequencies and positions designed for the nt821(del11) and F94L polymorphisms identification in the cattle MSTN gene. The results of the sequencies alignment of Bos taurus myostatin gene were shortly described in the text.

The data from Figure 7 were moved into Table 4. Figure 7 was deleted.

It should be note, line 685 in the manuscript is absent because the manuscript consists of fewer lines. We dare to assume that this comment referred to the last paragraph of the conclusion section. This conclusion seems logical for us. The CRISPR/CAS9 technology proposed the using of the high technological methods of the genome edition is undouble very perspective but it is enough complicated and expensive. The achievement of the breeding goals by gene selection could be a possibility to use the easier and cheaper methods leading to the achievement the same goal and, respectively, increasing the profitability of the beef cattle area. 

We really hope that the changes made with your help will improve the quality of the manuscript and it will be worthy of publication in a special issue dedicated to the myostatin gene of the Animals journal.

Round 3

Reviewer 3 Report

Dear Authors,

I would suggest that send your manuscript to a professional English editing service to check your academic manuscript and make sure it is well-written as some sentences are still unclear.

Thanks